# RoPK: A Head-Level Key Cache Channel Pruning Method for Efficient Long-Context LLM Inference

## Abstract

The substantial memory overhead of the Key-Value (KV) cache is a critical bottleneck in Large Language Models (LLMs), limiting context length and inference throughput. While prior compression techniques have targeted various dimensions of the cache, they typically assume a uniform channel allocation for all attention heads, which leads to inaccurate channel pruning and a significant drop in accuracy. This paper introduces a novel head-level key cache channel pruning method that allocates channel budgets based on a new head importance estimation algorithm derived from Rotary Position Embedding (RoPE). We posit that an effective attention head should leverage different RoPE frequencies to capture dependencies at corresponding distances—low frequencies for long-range and high frequencies for short-range. Consequently, we define head importance as the correlation score between the attention contribution of frequency dimensions and the relative textual distance. Based on this metric, our method allocates larger channel budgets to more important heads. Within each head, channels with lower attention contribution scores are pruned. This approach enables significant, non-uniform channel pruning in the key cache without sacrificing performance. Evaluations on LLaMA and Mistral models using LongBench, RULER, and Needle-in-a-Haystack benchmarks demonstrate the method's effectiveness. With 60% of the channels pruned, our method achieves almost no performance degradation on LongBench. At a more aggressive pruning rate of 70%, it surpasses the state-of-the-art Think method by 2%. Code is available in the supplementary material.

## 1 Introduction

The advent of Large Language Models (LLMs) (Achiam et al., 2023; Guo et al., 2025; Yang et al., 2025; Team et al., 2025) has marked a new epoch in natural language processing, delivering unprecedented capabilities, especially in comprehending and generating long-context text (Team et al., 2024; Naveed et al., 2025; Liu et al., 2025a). A foundational technology for their efficient autoregressive inference is the Key-Value (KV) cache, which stores the key and value projections of processed tokens to prevent redundant computation (Vaswani et al., 2017). However, this efficiency comes at a steep price: the KV cache's memory footprint scales linearly with sequence length and batch size, often surpassing the memory required for the model parameters themselves (Pope et al., 2023). This memory overhead represents a critical bottleneck, constraining maximum context lengths, reducing inference throughput, and impeding the practical deployment of these powerful models (Zhou et al., 2024; Liu et al., 2025b).

To address this challenge, a spectrum of KV cache compression techniques has emerged, each targeting a different dimension of the cache tensor ($B \times S \times L \times H \times D$) (Singhania et al., 2024; Kang et al., 2024; Chen et al., 2025; Sun et al., 2025). To manage the sequence length dimension ($S$), eviction strategies propose discarding the KV cache of less important tokens (Zhang et al., 2023; Xiao et al., 2024; Li et al., 2024c; Tang et al., 2024; Xiao et al., 2025). To reduce the number of layers ($L$), others have explored cross-layer KV sharing or merging (Liu et al., 2024; Yang et al., 2024b). Along the head dimension ($H$), methods allocate the cache budget based on head importance, retaining more tokens for critical heads (Feng et al., 2024; Fu et al., 2025). The channel dimension ($D$) has remained comparatively underexplored. The recent Think method (Xu et al., 2025) made

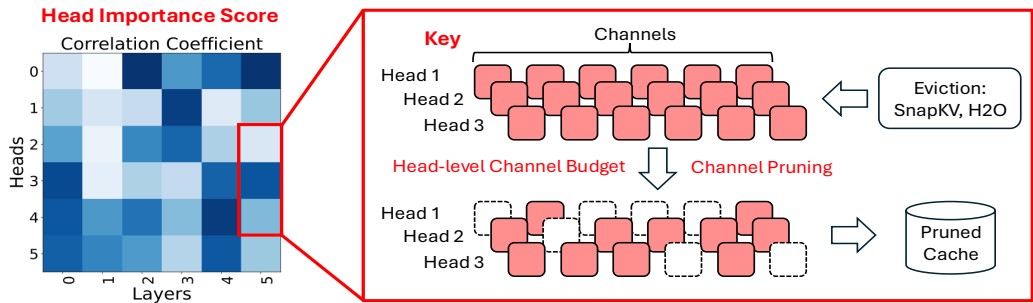

Figure 1: Our proposed RoPK pruning framework. We use the correlation between frequency contribution and relative distance as a score to assess the importance of attention heads. Attention heads with stronger positive correlation are considered more important and are allocated a larger channel budget. Subsequently, we prune the channels with lower attention contribution scores according to the channel budget within each head.

initial strides by pruning key cache channels, using a query-dependent criterion to assess channel importance.

Despite these advances, a crucial assumption persists: all attention heads are treated uniformly in terms of channel allocation. No existing work has explored assigning a variable number of channels to different heads based on their functional importance. This paper challenges this uniform allocation assumption. We introduce a novel, frequency-aware channel pruning methodology that allocates channel budgets dynamically across different attention heads. Our approach is guided by a new algorithm for determining head importance, derived from the principles of Rotary Position Embedding (RoPE) (Su et al., 2024). Heads deemed more critical are allocated a larger channel budget, while less important heads are pruned more aggressively, enabling significant cache reduction without compromising performance.

Our approach is rooted in a novel interpretation of the functional roles of different frequency dimensions within RoPE. We observe that RoPE's high-frequency rotations are inherently better suited for modeling short-range dependencies, while its low-frequency rotations excel at capturing long-range relationships. We posit that a well-performing attention head should leverage this specialization effectively: its high-frequency dimensions should encode short-distance information, and its low-frequency dimensions should encode long-distance information. Based on this insight, we propose a new head importance metric: the correlation between the contribution of frequency dimensions and the relative distance of tokens. Heads exhibiting a strong positive correlation are considered more specialized and thus more important, warranting a higher channel budget. Within each head, we then prune the least important channel pairs based on their attention contribution scores.

To validate our approach, we conduct extensive experiments on leading open-source models, including the LLaMA-3 (Meta, 2024) and Mistral (Jiang et al., 2023) series. We evaluate performance on a suite of long-context benchmarks, including LongBench (Bai et al., 2024a), RULER (Hsieh et al., 2024), and Needle-in-a-Haystack (Kamradt, 2023). The results demonstrate that our method enables substantial KV cache compression. For instance, with a 60% channel pruning rate, our method shows almost no performance degradation on LongBench. At a more aggressive pruning rate of 70%, it outperforms the state-of-the-art Think method by 2%.

## 2 RELATED WORK

**KV Cache Compression.** Many KV cache compression methods have been proposed to reduce the high memory cost associated with KV cache in LLMs. One major approach involves reducing the cache size's linear growth with sequence length by implementing eviction strategies that discard less important token caches (Ge et al., 2024; Liu et al., 2023; Chen et al., 2025; Adnan et al., 2024; Zhao et al., 2024a). Other methods optimize along the layer dimension, reducing the number of caches by enabling cross-layer sharing or merging (Zhang et al., 2024c; Kim et al., 2024; Nawrot et al., 2024). Additionally, optimization in the head dimension involves allocating the KV cache budget based on the importance of different attention heads (Zhang et al., 2024b; Fu et al., 2025; Feng et al., 2025).

A less explored area is the channel dimension, where techniques like pruning key cache channels based on query-dependent importance have been proposed (Xu et al., 2025). In addition, we can also compress the KV cache using quantization (Duanmu et al., 2024; Kang et al., 2024; Yang et al., 2024a) and low-rank decomposition techniques (Chang et al., 2025; Sun et al., 2025). Different from the above methods, our approach is the first to propose a head-level channel pruning method that dynamically allocates the channel budget across different attention heads.

**Structured Pruning of LLMs.** Structured pruning is a model compression method that makes LLMs smaller and faster by removing entire structural components like neurons (Ma et al., 2023; Ashkboos et al., 2024; Li et al., 2024a), attention heads (Xia et al., 2024; Zhang et al., 2024a; An et al., 2024), or layers (Song et al., 2024; Men et al., 2025). However, these methods require modifying the model structure of LLMs, which leads to a significant degradation in model performance (Wang et al., 2024). In contrast, our method does not need to alter the model structure. Instead, it reduces the memory footprint of LLMs by pruning the unimportant channels in the KV cache (Xu et al., 2025), achieving efficient pruning of LLMs while maintaining nearly lossless accuracy.

**Rotary Positional Encoding (RoPE).** RoPE (Su et al., 2024) enhances Transformer models by encoding positional information through rotational transformations applied to query and key vectors in self-attention and has been adopted by more and more LLMs (Zhao et al., 2024b; Li et al., 2024b; Minaee et al., 2024). It operates by splitting the embedding dimensions into pairs and applying rotations at different frequencies, which makes different frequencies contribute differently to the model's ability to capture contextual dependencies (Fang et al., 2024; Peng et al., 2024). Barbero et al. (Barbero et al., 2025) proposed that high frequencies in RoPE are used for positional attention, while low frequencies are used for semantic attention. Chiang et al. (Chiang & Yogatama, 2025) found that the dimensions corresponding to high frequencies in RoPE may be inefficient, and these dimensions do not help the model in long context tasks. In this paper, we propose a novel head importance metric based on RoPE and prune channels within each head according to the attention contribution scores of the frequency pairs.

# 3 BACKGROUND: ROTARY POSITIONAL ENCODING (RoPE)

**Notation.** We use boldface letters denote vectors (*e.g.,* $\boldsymbol{x}, \boldsymbol{y}$) and matrices (*e.g.,* $\boldsymbol{X}, \boldsymbol{Y}$).

## 3.1 DEFINITION

RoPE (Su et al., 2024) is a technique to encode positional information by rotating the query and key vectors based on their absolute position before the attention scores are computed. The primary objective is to formulate the transformation such that the inner product between a query vector at position $m$ and a key vector at position $n$ depends solely on their relative position $m - n$.

Let $\boldsymbol{q}_m$ and $\boldsymbol{k}_n$ be the original query and key vectors at positions $m$ and $n$, respectively. RoPE operates by partitioning the $D$-dimensional embedding into $D/2$ two-dimensional sub-vectors and applying a rotation to each. For any given 2D sub-vector $\boldsymbol{x} = [x_1, x_2]^\top$, its rotation by an angle $\alpha$ is performed by multiplying it with a rotation matrix:

$$f(\boldsymbol{x}, \alpha) = \begin{pmatrix} \cos\alpha & -\sin\alpha \\ \sin\alpha & \cos\alpha \end{pmatrix} \begin{pmatrix} x_1 \\ x_2 \end{pmatrix} \tag{1}$$

For a $D$-dimensional vector, this rotation is applied to each of the $D/2$ pairs of dimensions. The angle of rotation for the $i$-th pair $(x_{2i}, x_{2i+1})$ at a given position $t$ is set to $t\theta_i$. The frequencies $\theta_i$ are predefined constants: $\theta_i = base^{-2i/D}$ for $i \in \{0, \dots, D/2 - 1\}$, where $base$ is a chosen base, which by default is 10,000 (Su et al., 2024).

In the context of a large language model's attention mechanism, this transformation is applied to the query vector $\boldsymbol{q}_m$ and the key vector $\boldsymbol{k}_n$. This can be represented by a block-diagonal rotation matrix $\boldsymbol{R}_{\Theta,t}$ for any position $t$, where:

$$\boldsymbol{R}_{\Theta,t} = \text{diag}\left( \begin{pmatrix} \cos(t\theta_0) & -\sin(t\theta_0) \\ \sin(t\theta_0) & \cos(t\theta_0) \end{pmatrix}, \cdots, \begin{pmatrix} \cos(t\theta_{D/2-1}) & -\sin(t\theta_{D/2-1}) \\ \sin(t\theta_{D/2-1}) & \cos(t\theta_{D/2-1}) \end{pmatrix} \right) \tag{2}$$

The transformed query and key vectors, $\boldsymbol{q}'_m$ and $\boldsymbol{k}'_n$, are thus $\boldsymbol{q}'_m = \boldsymbol{R}_{\Theta,m}\boldsymbol{q}_m$ and $\boldsymbol{k}'_n = \boldsymbol{R}_{\Theta,n}\boldsymbol{k}_n$. Given that rotation matrices are orthogonal, it holds that $\boldsymbol{R}^\top_{\Theta,m}\boldsymbol{R}_{\Theta,n} = \boldsymbol{R}_{\Theta,n-m}$. The inner product is therefore:

$$(\boldsymbol{q}'_m)^\top \boldsymbol{k}'_n = (\boldsymbol{R}_{\Theta,m}\boldsymbol{q}_m)^\top (\boldsymbol{R}_{\Theta,n}\boldsymbol{k}_n) = \boldsymbol{q}^\top_m \boldsymbol{R}^\top_{\Theta,m}\boldsymbol{R}_{\Theta,n}\boldsymbol{k}_n = \boldsymbol{q}^\top_m \boldsymbol{R}_{\Theta,n-m}\boldsymbol{k}_n \tag{3}$$

The above result shows that the interaction between the query and key vectors is dependent on their original vector values and their relative displacement $n - m$, thus successfully encoding relative positional information into the self-attention mechanism. These rotated query and key vectors are then used to calculate the attention scores.

## 3.2 FREQUENCY CONTRIBUTION IN ROPE

RoPE achieves the rotation of key and value vectors by applying different frequencies to different dimensions. To analyze the contribution of different frequency components within RoPE to the overall attention score, we decompose the inner product $(\boldsymbol{q}'_m)^\top \boldsymbol{k}'_n$ into terms corresponding to each dimension. Let $a_i$ denote the contribution from the $i$-th dimension:

$$a_i = q'_{m,i} k'_{n,i} \tag{4}$$

The total inner product is then the sum of these terms: $(\boldsymbol{q}'_m)^T \boldsymbol{k}'_n = \sum_{i=0}^{D-1} a_i$. Given that RoPE operates on pairs of dimensions $(2i, 2i + 1)$ by applying a rotation with the same frequency $\theta_i$, we group the contributions accordingly. We define the contribution from the $i$-th pair as the sum of the contributions from its constituent dimensions:

$$c_i = a_{2i} + a_{2i+1} \quad \text{for } i = 0, 1, \ldots, \frac{D}{2} - 1 \tag{5}$$

The term $c_i$ thus represents the total contribution to the attention score from the dimension pair associated with frequency $\theta_i$. By expanding this definition using the RoPE transformation, we can express $c_i$ as a function of the original query and key vectors $(\boldsymbol{q}_m, \boldsymbol{k}_n)$ and their relative position $m - n$:

$$c_i = (q_{m,2i}k_{n,2i} + q_{m,2i+1}k_{n,2i+1})\cos((m-n)\theta_i) + (q_{m,2i}k_{n,2i+1} - q_{m,2i+1}k_{n,2i})\sin((m-n)\theta_i) \tag{6}$$

The full dot product of the transformed vectors is the sum of these individual frequency contributions:

$$(\boldsymbol{q}'_m)^\top \boldsymbol{k}'_n = \sum_{i=0}^{D/2-1} c_i \tag{7}$$

Therefore, the value $c_i$ serves as a metric to measure the contribution of each frequency component $\theta_i$ to the final attention score, enabling a detailed analysis of how different frequencies in RoPE contribute to semantic modeling.

## 4 METHODOLOGY

In this section, we first examine some theoretical properties of frequency contribution in RoPE. Then, based on these theoretical properties, we propose a metric to determine the importance of attention heads, and develop a head-level pruning rate allocation algorithm according to head importance. Finally, we put forward a pruning algorithm for the channels within each head.

## 4.1 PROPERTIES OF DIFFERENT FREQUENCY IN ROPE

We begin by establishing a key property that links the rotational frequencies of RoPE to the sensitivity of their corresponding contributions to changes in relative position.

**Theorem 1.** *The maximum rate of change of the frequency contribution $c_i(t)$ with respect to the relative position $t$ is directly proportional to its corresponding frequency $\theta_i$. Formally:*

$$\max_t \left| \frac{dc_i(t)}{dt} \right| \propto \theta_i \tag{8}$$

*where $c_i(t)$ is the contribution from the $i$-th frequency as defined in Equation 6.*

A detailed proof is provided in Appendix B. Theorem 1 establishes that the contribution term $c_i$ associated with a higher frequency $\theta_i$ is inherently more sensitive to small changes in relative position $t$. Conversely, contributions from lower frequencies exhibit a slower rate of change, making them less sensitive to local positional shifts. This inherent property leads to a natural specialization of the frequency components, which we formalize in the following corollary.

**Corollary 1.** *The high-frequency components of RoPE are better suited for encoding short-range dependencies. Conversely, the low-frequency components are better suited for encoding long-range dependencies.*

The proof is available in Appendix C. Building on this corollary, which establishes a clear division of labor between high and low frequencies, we can now develop a metric to evaluate the importance of an attention head based on its frequency utilization.

### 4.2 HEAD IMPORTANCE SCORE

To quantify how different attention heads utilize the RoPE frequency spectrum, we propose a novel metric based on their specialization for short- and long-range dependencies. For a given query-key pair $(\boldsymbol{q}_m, \boldsymbol{k}_n)$ with relative distance $t = m - n$, we first compute the contribution $c_i(t)$ from each frequency pair $i \in \{0, \dots, D/2 - 1\}$ using Equation 6. We partition the frequency pairs into "high-frequency" ($i < D/4$) and "low-frequency" ($i \geq D/4$) sets. We then define the **low-frequency contribution ratio**, $F_t$, as the fraction of the total attention score that originates from the low-frequency components:

$$F_t = \frac{\sum_{i=D/4}^{D/2-1} c_i(t)}{\sum_{i=0}^{D/2-1} c_i(t)} \tag{9}$$

A high value of $F_t$ indicates that the attention score for that relative distance $t$ is dominated by low-frequency contributions. Conversely, a low value of $F_t$ signifies dominance by high-frequency contributions.

For each attention head, we compute $F_t$ across a sample of query-key pairs with varying relative distances $t$. We then quantify the relationship between positional range and frequency usage by calculating the Spearman's rank correlation coefficient (Sedgwick, 2014) between the set of relative distances $\{t\}$ and their corresponding low-frequency contribution ratios $\{F_t\}$. A high positive correlation (approaching +1) indicates that the head behaves in accordance with the theoretical specialization outlined in Corollary 1: it increasingly relies on low-frequency components for longer distances (large $t$) and high-frequency components for shorter distances (small $t$).

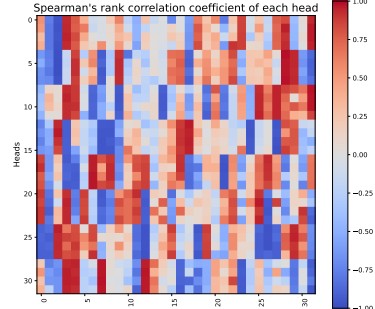

Figure 2: Spearman's rank correlation between the low-frequency contribution ratio ($F_t$) and relative distance ($t$) for all attention heads.

Figure 2 plots this correlation score for every attention head in the LLaMA-3-8B-Instruct model. As shown, the scores exhibit significant variance, with many heads having scores near +1, while others have scores near 0 or even negative values, suggesting a lack of specialization.

To validate that this correlation score reflects head importance, we conduct an ablation study. We identify the top 10% of heads with the highest correlation scores and, as a control, the bottom 10% with the lowest scores. We then measure model performance on several long-context question-answering tasks from the LongBench dataset after zeroing out the key-value cache for the selected heads during inference.

The results are presented in Table 1. Masking the top 10% of heads causes a substantial degradation in performance, with an average accuracy drop of 8.4%. In contrast, masking the bottom 10% results in a much smaller performance drop of 3.59%. This experiment provides strong evidence that heads with a high correlation score, those that effectively specialize their

Table 1: Ablation of attention heads based on their correlation score.

| Masked | HotpotQA | 2WikiMQA | Musique | Average |
|--------|----------|----------|---------|---------|
| None | 42.75 | 34.55 | 20.87 | **32.73** |
| Top 10% | 32.48 | 25.07 | 15.43 | **24.33** |
| Bottom 10% | 39.17 | 30.19 | 18.06 | **29.14** |

frequency usage according to relative distance, are critical for the model's performance on tasks requiring long-context reasoning. We therefore adopt this correlation score as a robust metric for head importance. In the following section, we leverage this importance score to allocate the pruning rate to each attention head.

## 4.3 HEAD-LEVEL PRUNING RATE ALLOCATION

We first use the method proposed in Section 4.2 to calculate the importance score of each head. The set of importance scores for all heads is $\boldsymbol{s} = \{s_1, s_2, \ldots, s_H\}$, where $H$ represent the number of heads in each layer. Then we use the following formula to derive the pruning rate $p_k$ for each head:

$$s'_k = \left( \frac{s_k - \min(\boldsymbol{s})}{\max(\boldsymbol{s}) - \min(\boldsymbol{s})} \right) \times 2\beta, \quad p_k = 1 - [s'_k - \text{mean}(\boldsymbol{s'}) + (1 - P)] \tag{10}$$

where $\max(\cdot)$, $\min(\cdot)$, and $\text{mean}(\cdot)$ represent the maximum value, minimum value, and mean value of the set, respectively. $P$ is a pre-defined average pruning rate (the proportion of pruned channels to the total number of channels) for all heads, and $\beta$ is a hyperparameter that controls the difference in pruning rates between different heads. A larger $\beta$ value leads to a greater difference in pruning rates among different heads.

## 4.4 FREQUENCY-AWARE CHANNEL PRUNING

After determining the target pruning rate $p_k$ for each head, we must select which specific channels to remove. Given that RoPE's rotational transformations operate on pairs of dimensions $(2i, 2i + 1)$, we adopt a pair-wise pruning strategy to preserve the integrity of the positional encoding mechanism.

For each frequency pair $i \in \{0, 1, \ldots, D/2 - 1\}$, we first compute its contribution $c_i(m, n)$ for every query-key position pair $(m, n)$ using Equation 6. These contributions form a matrix $\boldsymbol{C}_i \in \mathbb{R}^{S \times S}$, where $(\boldsymbol{C}_i)_{mn} = c_i(m, n)$, $S$ is the sequence length, and $D$ is the head dimension. The importance score for the $i$-th frequency pair, denoted $\text{imp}(i)$, is then defined as the squared Frobenius norm of its contribution matrix. This score aggregates the magnitude of the pair's influence over the entire sequence:

$$\text{imp}(i) = ||\boldsymbol{C}_i||_F^2 = \sum_{m=1}^{S} \sum_{n=1}^{S} (c_i(m, n))^2 \tag{11}$$

This metric quantifies the overall impact of the $i$-th frequency component $\theta_i$ on the attention mechanism for the given input. A higher value of $\text{imp}(i)$ indicates greater importance.

With these importance scores, the pruning procedure is as follows. First, we calculate the number of channel pairs to retain for head $k$: $N_{\text{keep}} = \lfloor (1 - p_k) \frac{D}{2} \rfloor$. We then identify the set $\mathcal{I}_k$ containing the indices of the $N_{\text{keep}}$ frequency pairs with the highest importance scores. Finally, we construct a diagonal binary pruning mask $\boldsymbol{M}_k \in \{0, 1\}^{D \times D}$ that preserves the channels corresponding to the indices in $\mathcal{I}_k$. The diagonal elements of the mask are defined as:

$$(\boldsymbol{M}_k)_{j,j} = \begin{cases} 1 & \text{if } \lfloor j/2 \rfloor \in \mathcal{I}_k \\ 0 & \text{otherwise} \end{cases} \quad \text{for } j = 0, 1, \ldots, D - 1 \tag{12}$$

For the $k$-th head, the query and key can be denoted as $\boldsymbol{Q}_k \in \mathbb{R}^{S \times D}$ and $\boldsymbol{K}_k \in \mathbb{R}^{S \times D}$ respectively. After applying this mask, the pruned attention computation becomes $(\boldsymbol{Q}_k \boldsymbol{M}_k)(\boldsymbol{K}_k \boldsymbol{M}_k)^\top$. This ensures that channels are pruned symmetrically from both the query and key projection heads, maintaining the paired structure of RoPE.

## 5 EXPERIMENTS

### 5.1 EXPERIMENTAL SETUP

**Models.** We validated the effectiveness of our method on widely used open-source LLMs, including LLaMA-3-8B-Instruct (Meta, 2024) and Mistral-7B-Instruct-v0.2 (Jiang et al., 2023).

Table 2: Experiment results of LLaMA-3-8B-Instruct and Mistral-7B-Instruct-v0.2 models on Longbench benchmark at different KV sizes and pruning rates. Think($p$) and RoPK($p$) denote adopting a pruning rate of $p$ for the key cache channels.

| Method | Single-Document QA | | | Multi-Document QA | | | Summarization | | | Few-shot Learning | | | Synthetic | | Code | | Avg. |
|---|---|---|---|---|---|---|---|---|---|---|---|---|---|---|---|---|---|
| | NrtvQA | Qasper | MF-en | HotpotQA | 2WikiMQA | Musique | GovReport | QMSum | MultiNews | TREC | TriviaQA | SAMSum | PCount | PRe | Lcc | RB-P | |
| *LLaMA-3-8B-Instruct, KV-size Full* | | | | | | | | | | | | | | | | | |
| ALL KV | 25.56 | 32.27 | 39.71 | 43.56 | 35.09 | 21.18 | 28.71 | 23.26 | 26.64 | 73.50 | 90.48 | 42.33 | 4.80 | 69.25 | 59.29 | 54.05 | 41.86 |
| *LLaMA-3-8B-Instruct, KV-size 128* | | | | | | | | | | | | | | | | | |
| SnapKV | 21.19 | 13.55 | 32.64 | 38.75 | 29.64 | 18.73 | 18.98 | 21.62 | 20.26 | 45.00 | 88.36 | 37.64 | 5.13 | 68.85 | 55.84 | 51.82 | 35.50 |
| +Think(0.6) | 22.00 | 12.62 | 33.52 | 37.04 | 28.78 | 18.09 | 18.09 | 21.23 | 18.93 | 39.50 | 86.53 | 37.23 | 5.59 | 69.20 | 55.24 | 56.35 | 35.00 |
| **+RoPK(0.6)** | 21.39 | 12.94 | 35.83 | 39.07 | 29.39 | 17.66 | 18.53 | 21.36 | 19.00 | 41.00 | 87.00 | 36.56 | 5.67 | 69.50 | 54.63 | 56.81 | **35.39** |
| +Think(0.7) | 20.70 | 11.14 | 32.73 | 35.10 | 28.42 | 17.38 | 17.31 | 20.77 | 17.83 | 37.50 | 86.84 | 35.99 | 5.92 | 69.50 | 52.93 | 54.66 | 34.05 |
| **+RoPK(0.7)** | 21.13 | 12.52 | 34.50 | 36.67 | 28.59 | 17.74 | 18.04 | 20.76 | 18.40 | 38.50 | 87.61 | 35.66 | 5.67 | 69.50 | 52.62 | 55.01 | **34.56** |
| *LLaMA-3-8B-Instruct, KV-size 512* | | | | | | | | | | | | | | | | | |
| SnapKV | 24.84 | 23.96 | 38.77 | 42.75 | 34.55 | 20.87 | 22.26 | 22.61 | 23.97 | 70.00 | 90.52 | 40.29 | 5.81 | 69.50 | 59.04 | 51.81 | 40.10 |
| +Think(0.6) | 25.76 | 22.77 | 38.37 | 40.44 | 32.91 | 19.90 | 20.84 | 22.21 | 22.55 | 59.00 | 90.32 | 37.80 | 6.31 | 69.20 | 58.76 | 58.18 | 39.08 |
| **+RoPK(0.6)** | 24.83 | 24.23 | 38.86 | 41.06 | 32.46 | 19.98 | 20.89 | 22.45 | 23.33 | 61.00 | 90.36 | 37.69 | 6.17 | 68.95 | 58.25 | 58.28 | **39.30** |
| +Think(0.7) | 23.66 | 14.97 | 36.11 | 38.56 | 30.28 | 19.70 | 19.27 | 22.10 | 19.79 | 43.00 | 87.81 | 35.20 | 6.00 | 69.30 | 54.69 | 55.67 | 36.01 |
| **+RoPK(0.7)** | 24.95 | 22.01 | 39.51 | 38.49 | 32.33 | 19.82 | 19.39 | 22.00 | 21.78 | 51.00 | 90.62 | 36.02 | 6.13 | 69.35 | 53.94 | 56.06 | **37.71** |
| *LLaMA-3-8B-Instruct, KV-size 1024* | | | | | | | | | | | | | | | | | |
| SnapKV | 24.62 | 25.99 | 37.64 | 43.84 | 34.99 | 20.00 | 24.28 | 22.39 | 25.63 | 72.50 | 90.56 | 40.41 | 5.36 | 69.25 | 60.57 | 56.11 | 40.88 |
| +Think(0.6) | 24.48 | 27.19 | 38.22 | 41.96 | 31.64 | 20.18 | 21.89 | 22.83 | 23.68 | 69.25 | 90.19 | 38.51 | 6.12 | 69.50 | 58.35 | 59.26 | 40.20 |
| **+RoPK(0.6)** | 25.01 | 26.77 | 39.51 | 42.22 | 32.86 | 19.62 | 21.90 | 22.91 | 24.31 | 69.50 | 90.23 | 37.98 | 6.23 | 69.55 | 58.42 | 59.29 | **40.39** |
| +Think(0.7) | 23.73 | 16.12 | 38.20 | 39.66 | 29.94 | 18.65 | 19.62 | 21.40 | 19.48 | 46.00 | 87.50 | 34.63 | 6.06 | 69.50 | 54.49 | 54.75 | 36.23 |
| **+RoPK(0.7)** | 23.19 | 23.94 | 40.07 | 37.83 | 31.52 | 19.10 | 20.12 | 22.49 | 21.96 | 56.50 | 90.16 | 35.68 | 6.07 | 69.50 | 54.19 | 55.90 | **38.01** |
| *LLaMA-3-8B-Instruct, KV-size 2048* | | | | | | | | | | | | | | | | | |
| SnapKV | 25.86 | 29.55 | 41.10 | 44.99 | 35.80 | 21.81 | 25.98 | 23.40 | 26.46 | 73.50 | 90.56 | 41.66 | 5.17 | 69.25 | 58.67 | 51.52 | 41.58 |
| +Think(0.6) | 24.95 | 28.91 | 40.44 | 41.30 | 29.99 | 21.05 | 23.24 | 22.90 | 24.99 | 72.12 | 90.36 | 38.50 | 5.71 | 69.50 | 59.77 | 59.20 | 40.81 |
| **+RoPK(0.6)** | 25.27 | 28.98 | 40.97 | 42.72 | 33.29 | 20.99 | 23.34 | 23.34 | 25.12 | 72.20 | 90.64 | 38.53 | 6.23 | 69.55 | 58.21 | 59.38 | **41.16** |
| +Think(0.7) | 24.87 | 17.41 | 37.78 | 39.59 | 30.57 | 17.99 | 19.77 | 22.04 | 19.86 | 50.50 | 88.03 | 34.57 | 6.06 | 69.25 | 54.09 | 55.38 | 36.74 |
| **+RoPK(0.7)** | 24.11 | 26.53 | 39.21 | 37.63 | 33.31 | 18.14 | 20.05 | 22.66 | 20.89 | 64.50 | 90.16 | 35.11 | 6.12 | 69.50 | 54.76 | 56.73 | **38.71** |
| *Mistral-7B-Instruct-v0.2, KV-size Full* | | | | | | | | | | | | | | | | | |
| ALL KV | 26.63 | 32.99 | 49.34 | 42.77 | 27.35 | 18.77 | 32.87 | 24.24 | 27.10 | 71.00 | 86.23 | 42.96 | 2.75 | 86.98 | 56.93 | 54.49 | 42.71 |
| *Mistral-7B-Instruct-v0.2, KV-size 512* | | | | | | | | | | | | | | | | | |
| SnapKV | 24.44 | 27.81 | 48.98 | 39.46 | 25.25 | 16.98 | 23.70 | 22.96 | 24.37 | 67.00 | 85.88 | 41.26 | 2.78 | 86.56 | 56.46 | 53.41 | 40.46 |
| +ThinK(0.7) | 23.72 | 25.79 | 47.98 | 37.25 | 24.24 | 17.19 | 21.99 | 23.30 | 22.99 | 63.00 | 85.28 | 36.28 | 3.25 | 77.53 | 52.61 | 50.32 | 38.30 |
| **+RoPK(0.7)** | 24.02 | 27.02 | 48.77 | 39.12 | 24.33 | 17.20 | 23.00 | 23.40 | 23.82 | 65.00 | 85.91 | 37.11 | 3.51 | 75.17 | 52.76 | 50.42 | **38.79** |

**Benchmarks.** We demonstrate the performance of our method on three widely recognized long-context understanding benchmarks for LLMs, namely LongBench (Bai et al., 2024a), Needle-in-a-Haystack (Kamradt, 2023) and RULER (Hsieh et al., 2024).

**Baselines.** We compare state-of-the-art KV cache compression baselines, including Heavy Hitter Oracle (H2O) (Zhang et al., 2023), SnapKV (Li et al., 2024c), and ThinK (Xu et al., 2025). Specifically, H2O, SnapKV are token eviction methods and Think is a key cache channel pruning method.

Detailed information about the above benchmarks and baselines can be found in Appendix D.

**Implementation Details.** All experiments are performed on A NVIDIA A100 80GB GPU. The hyperparameter $\beta$, which controls the difference in pruning rates among different attention heads, is selected from the set $\{0.02, 0.05, 0.1, 0.2\}$, and we report the best performance. Since computing the correlation coefficient between the low-frequency contribution ratio and the relative distance for all attention query-key pairs is time-consuming and query-key pairs that contribute less to the attention score tend to introduce noise when estimating the correlation coefficient, resulting in a decrease in estimation accuracy, we use the top 100 query-key pairs that contribute the most to the attention scores. Our evaluation follows the basic setup of the SnapKV (Sun et al., 2025) and Think (Xu et al., 2025) methods. To ensure a fair comparison between the baseline methods and ours, we adopt the same hyperparameters as the baselines. For the SnapKV and Think methods, we use a max-pooling kernel size of 7 and an observation window size of 32.

## 5.2 RESULTS ON LONGBENCH

**Quantitative Evaluation.** We present the LongBench results on LLaMA-3-8B-Instruct and Mistral-7B-Instruct-v0.2 models of our proposed key cache channel pruning method integrated with the SnapKV method in Table 2. Specifically, we further pruned unimportant key cache channels based

on the SnapKV method. From the experimental data in the table, we can observe that our method successfully prunes the channels of the key cache, while having minimal impact on the accuracy of SnapKV. This indicates that existing key caches have room for further reduction in the channel dimension, and our method can further decrease the key cache size by channel pruning. Additionally, under different KV size and pruning rate settings, the accuracy of our method is all superior to that of the Think method and the accuracy of our RoPK method is significantly better than that of the Think method under the setting of a 70% pruning rate. Our method further pushes the performance limits of key cache channel pruning. This demonstrates the superiority and robustness of our method.

**Orthogonality with Other Methods.** Notably, our method can be combined with other KV cache compression techniques to further enhance the compression effect. We present the LongBench benchmark results of our method combined with another KV cache eviction method H2O (Zhang et al., 2023) when applied to the LLaMA-3-8B-Instruct model and set KV size to 512. Specifically, we further prune the channels of the key cache based on H2O. The experimental results are shown in Table 3. Our method achieves a further reduction in KV cache memory while causing only minimal performance loss when combined with H2O. Meanwhile, the performance of our method is still better than that of Think method.

Table 3: Experimental results of LLaMA-3-8B-Instruct model on LongBench benchmark when combined with H2O. Think($p$) and RoPK($p$) denote adopting a pruning rate of $p$ for the key cache channels.

| Method | Single-Document QA | | | Multi-Document QA | | | Summarization | | | Few-shot Learning | | | Synthetic | | Code | | Avg. |
| | NrtvQA | Qasper | MF-en | HotpotQA | 2WikiMQA | Musique | GovReport | QMSum | MultiNews | TREC | TriviaQA | SAMSum | PCount | PRe | Lcc | RB-P | |
|---|---|---|---|---|---|---|---|---|---|---|---|---|---|---|---|---|---|
| H2O | 23.52 | 17.93 | 34.68 | 42.11 | 33.52 | 19.92 | 22.11 | 22.56 | 23.82 | 41.00 | 90.46 | 40.20 | 5.87 | 69.50 | 56.71 | 51.69 | 37.23 |
| +ThinK(0.7) | 23.29 | 12.59 | 30.96 | 37.83 | 30.35 | 18.60 | 19.44 | 21.95 | 19.06 | 38.50 | 87.49 | 35.41 | 5.74 | 69.50 | 54.89 | 54.32 | 35.00 |
| **+RoPK(0.7)** | 22.73 | 14.04 | 34.54 | 39.22 | 30.13 | 18.32 | 19.47 | 22.62 | 20.94 | 39.00 | 89.77 | 35.73 | 6.22 | 69.50 | 53.88 | 54.75 | **35.68** |

## 5.3 RESULTS ON NEEDLE-IN-A-HAYSTACK

We present the experimental results of the pruned Mistral-7B-Instruct-v0.2 model on the Needle-in-a-Haystack test (Kamradt, 2023) in Table 4. Specifically, we show the experimental results for KV sizes ranging from 128 to 2048 and pruning rate is set to 60%, and we compare our method with the SnapKV and Think methods. The experimental data indicates that under most settings of KV sizes and pruning rates, the accuracy of our method is comparable to that of SnapKV and outperforms that of the Think method. In addition, we visualize the accuracy of the Needle-in-a-Haystack test under different token lengths and depths in Appendix E.

Table 4: Experimental results on Needles-in-a-Haystack test.

| Method | KVsize | | | |
| | 128 | 512 | 1024 | 2048 |
|---|---|---|---|---|
| SnapKV | 84.8 | 97.0 | 98.0 | 99.5 |
| +ThinK | 83.2 | 96.2 | 97.9 | 99.5 |
| **+RoPK** | **83.4** | **96.4** | **98.0** | **99.5** |

## 5.4 RESULTS ON RULER

We further present the accuracy of the pruned LLaMA-3-8B-Instruct model on the RULER benchmark in Table 5, with the input length set to 4K and the KV size is set to 2048. From the data in table, we can observe that the accuracy of our method is basically on par with that of SnapKV, and outperforms that of the Think method. Therefore, we conclude that further pruning of key cache channels does not affect the model's ability to extract key contextual information. Our method maintains superior performance even with further compression of the KV cache.

## 5.5 MORE EXPERIMENTAL RESULTS

We present present the experimental results of directly performing key cache channel pruning on the model in Appendix F. Our method can be directly applied to channel pruning of LLMs and achieves excellent performance.

Table 5: Experiment results of LLaMA-3-8B-Instruct model on RULER benchmark. Think($p$) and RoPK($p$) denote adopting a pruning rate of $p$ for the key cache channels.

| Method | Niah1 | Niah2 | MKey1 | MKey2 | MQuery | MValue | CWE | FWE | VT | Avg. |
|---|---|---|---|---|---|---|---|---|---|---|
| ALL KV | 100.00 | 100.00 | 99.40 | 100.00 | 99.85 | 98.85 | 99.76 | 92.00 | 99.40 | 98.81 |
| SnapKV | 100.00 | 100.00 | 99.40 | 100.00 | 99.90 | 98.90 | 99.56 | 89.40 | 99.24 | 98.49 |
| +ThinK(0.5) | 100.00 | 100.00 | 99.00 | 99.60 | 97.50 | 96.10 | 99.16 | 88.20 | 97.08 | 97.40 |
| **+RoPK(0.5)** | 100.00 | 100.00 | 99.60 | 99.60 | 98.65 | 95.90 | 99.26 | 88.33 | 98.44 | **97.75** |

## 5.6 ABLATION STUDIES

**Head-level Pruning Rate Allocation.** We show the impact of different $\beta$ values on the final model accuracy when combined with SnapKV under the setting of KV size 512 and a pruning rate of 70% in Figure 3. We report the average accuracy on the LongBench benchmark dataset of LLaMA-3-8B-Instruct model. The non-uniform pruning settings with different $\beta$ values outperform the uniform pruning rate setting ($\beta$=0). The above experimental results demonstrate the effectiveness of our head-level pruning rate allocation algorithm, which can assign reasonable channel budgets to different heads, and this significantly improves the model accuracy.

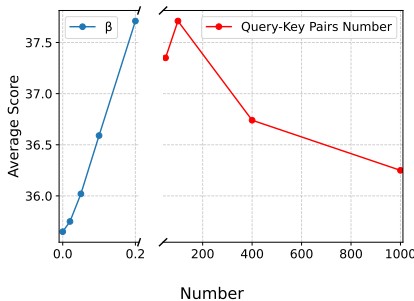

Figure 3: The effect of hyperparameter $\beta$ and the number of query–key pairs on the model's accuracy.

**Hyperparameter $\beta$.** We present the impact of different hyperparameter $\beta$ settings on model accuracy in Figure 3. A larger $\beta$ value indicates a greater difference in the number of pruned channels among different heads. The results show that the model achieves better performance under larger $\beta$ settings, which demonstrates that there are significant differences in the importance levels among different heads, and it is necessary to assign reasonable pruning rates to different heads.

**Number of Query-Key Pairs.** We present the impact of selecting different numbers of query-key pairs for estimating the correlation coefficient between the low-frequency contribution ratio and the relative distance on the final model accuracy in Figure 3. The results reported here are the average accuracy of the LLaMA-3-8B-Instruct model on the LongBench benchmark, when our method is combined with SnapKV, with the KV size set to 512 and a pruning rate of 70%. We report the final model accuracy when the top 50, 100, 400, and 1000 query-key pairs are selected. We found that the model achieves the highest accuracy when selecting the top 100 query-key pairs. This is because Spearman's rank correlation coefficient is prone to noise interference and yields unstable results when the sample size is extremely small. However, when increasing to 400, 1000, or even higher, query-key pairs with gradually decreasing attention scores are included. These pairs have weaker and more ambiguous semantic correlations, and the relationship between relative distance and frequency usage may become highly noisy, which further reduces the accuracy of estimating the true correlation. Therefore, we select the top 100 query-key pairs for estimating the correlation coefficient.

## 6 CONCLUSION

This paper introduces a frequency-aware channel pruning strategy that dynamically allocates KV cache budget across different attention heads. We propose a novel head importance metric rooted in the principles of RoPE. This metric evaluates how well a head specializes its frequency dimensions, measuring the correlation between the contribution of high/low frequencies and the relative distance of tokens. Heads showing a stronger positive correlation are considered more important and are allocated a larger channel budget. Meanwhile, we prune channels based on the attention contribution scores of frequency pairs within each head. Experimental results show that our method achieves efficient KV cache pruning while maintaining high performance in long-context tasks.

REPRODUCIBILITY STATEMENTS

The experimental setup is described in Section 5.1. We provide the code to reproduce our results in the supplementary materials, and our code will be released.

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

## APPENDIX

## A STATEMENT ON THE USE OF LARGE LANGUAGE MODELS

In the process of completing this paper, we used large language models (LLMs) to help us revise and polish the grammar, enhancing the clarity, grammatical accuracy, and textual standardization of the content.

## B PROOF OF THEOREM 1

*Proof.* The contribution $c_i(t)$ from the $i$-th frequency is given as a function of the relative position $t = m - n$:

$$c_i(t) = (q_{m,2i}k_{n,2i} + q_{m,2i+1}k_{n,2i+1})\cos(t\theta_i) + (q_{m,2i}k_{n,2i+1} - q_{m,2i+1}k_{n,2i})\sin(t\theta_i) \quad (13)$$

For simplicity, we can define coefficients $A_i$ and $B_i$ that are constant with respect to the relative position $t$:

$$A_i = q_{m,2i}k_{n,2i} + q_{m,2i+1}k_{n,2i+1}$$
$$B_i = q_{m,2i}k_{n,2i+1} - q_{m,2i+1}k_{n,2i} \tag{14}$$

Thus, the contribution can then be expressed as a sinusoidal function of $t$:

$$c_i(t) = A_i \cos(t\theta_i) + B_i \sin(t\theta_i) \tag{15}$$

We are interested in the sensitivity of $c_i(t)$ to changes in $t$, which is given by its derivative with respect to $t$:

$$\begin{aligned}
\frac{dc_i(t)}{dt} &= \frac{d}{dt}\left(A_i \cos(t\theta_i) + B_i \sin(t\theta_i)\right) \\
&= -A_i \sin(t\theta_i) \cdot \theta_i + B_i \cos(t\theta_i) \cdot \theta_i \\
&= \theta_i \left(B_i \cos(t\theta_i) - A_i \sin(t\theta_i)\right)
\end{aligned} \tag{16}$$

The magnitude of this derivative represents the rate of change of the contribution $c_i$ at a specific relative position $t$. The maximum magnitude of the derivative is:

$$\max_t \left| \frac{dc_i(t)}{dt} \right| = \max_t \left| \theta_i \left(B_i \cos(t\theta_i) - A_i \sin(t\theta_i)\right) \right| \tag{17}$$

Since $\theta_i > 0$, we have:

$$\max_t \left| \frac{dc_i(t)}{dt} \right| = \theta_i \cdot \max_t \left| B_i \cos(t\theta_i) - A_i \sin(t\theta_i) \right| = \theta_i \sqrt{A_i^2 + B_i^2} \tag{18}$$

The term $\sqrt{A_i^2 + B_i^2}$ depends only on the query and key vector, not on the frequency $\theta_i$. Therefore, the maximum magnitude of the derivative is directly proportional to its frequency $\theta_i$.

$$\max_t \left| \frac{dc_i(t)}{dt} \right| \propto \theta_i \tag{19}$$

This completes the proof. □

## C  PROOF OF COROLLARY 1

*Proof.* The corollary posits a functional specialization of RoPE's dimensions based on their associated frequencies. We prove this by examining the properties of high and low frequencies separately.

**Part 1: High Frequencies for Short-Range Dependencies**

The dimensions corresponding to small indices $i$ are associated with high frequencies, as $\theta_i = base^{-2i/d}$ is large when $i$ is small. From Theorem 1, a large $\theta_i$ implies a large value for $\max_t \left| \frac{dc_i(t)}{dt} \right|$. This indicates that the contribution $c_i(t)$ is highly sensitive to small perturbations in the relative position $t$.

The period of the sinusoidal function $c_i(t)$ is $T_i = 2\pi/\theta_i$. For high frequencies, this period is short, meaning $c_i(t)$ oscillates rapidly as $t$ changes. This rapid oscillation allows the model to assign substantially different values to $c_i(t)$ for small integer changes in $t$. This high-resolution signal is essential for distinguishing between nearby tokens. This property makes high-frequency components ideal for encoding short-range dependencies.

**Part 2: Low Frequencies for Long-Range Dependencies**

Conversely, dimensions corresponding to large indices $i$ are associated with low frequencies, as $\theta_i = base^{-2i/d}$ approaches zero when $i$ approaches $D/2 - 1$. According to Theorem 1, a small $\theta_i$ implies a small value for $\max_t \left| \frac{dc_i(t)}{dt} \right|$. This means the contribution $c_i(t)$ changes slowly with respect to $t$.

The period of oscillation, $T_i = 2\pi/\theta_i$, becomes very large for these low frequencies. Consequently, for small values of $t$, the function $c_i(t)$ is nearly constant, making it ineffective at distinguishing between adjacent positions. However, over a large range of $t$, the value of $c_i(t)$ varies smoothly and non-periodically (within the typical context window). This slow, gradual change provides a stable and unambiguous signal for coarse-grained relative distances. This property makes low-frequency components ideal for encoding long-range dependencies.

Thus, the inherent mathematical properties of RoPE's frequency spectrum lead to a natural division of labor, where high-frequency components are better suited for encoding short-range dependencies and low-frequency components are better suited for encoding long-range dependencies. This completes the proof. □

## D   DETAILED INFORMATION ABOUT THE BENCHMARKS AND BASELINES

**Benchmarks.**   LongBench (Bai et al., 2024b) is a comprehensive benchmark designed to evaluate the long-context understanding capabilities of LLMs. It includes 17 datasets covering 6 different tasks: single-document question answering, multi-document question answering, summarization, few-shot learning, synthetic tasks, and code completion. The average input length of LongBench is 6,711 words. The Needle-in-a-Haystack (Kamradt, 2023) test evaluates LLM's ability to retrieve a specific fact (the "needle") from a large body of text (the "haystack"). The test confirms if KV cache compression methods preserve the model's essential ability to recall specific details from extensive documents without performance degradation. RULER (Hsieh et al., 2024) is a synthetic benchmark to evaluate the real context capabilities of long-context language models. RULER consists of four main task categories, including retrieval, multi-hop tracing, aggregation, and question answering.

**Baselines.**   H2O (Zhang et al., 2023) proposes a KV cache eviction strategy, which achieves KV cache compression by dynamically retaining the "Heavy Hitters" (tokens that contribute the most to attention scores) and recent tokens. SnapKV (Li et al., 2024c) compresses the cache and accelerates long text generation by using an "observation window" to predict and retain the key KV cache that attention heads continuously focus on. ThinK (Xu et al., 2025) proposes a query-driven KV cache pruning method which prunes the least significant channels of key cache, thereby reducing memory usage while preserving model accuracy.

## E   VISUALIZATION RESULTS OF NEEDLE-IN-A-HAYSTACK

In addition to the results of Mistral-7B-Instruct-v0.2 on the Needle-in-a-Haystack task presented in Section 5.3, we also provide the visualization results of the Needle-in-a-Haystack test using the SnapKV, Think, and our RoPK methods in Figure 4, 5, 6 and 7. Compared with SnapKV, our RoPK method has a less impact on the ability to understand semantic information under different token limits and depths and outperforms that of the Think method.

## F   DIRECTLY PERFORMING KEY CACHE CHANNEL PRUNING ON LLM

In this section, we present the experimental results of directly applying our RoPK method to prune the key cache channels of LLM, without combining it with KV cache compression methods based on token eviction (*e.g.,* SnapKV and H2O). This means that the key cache of the LLM maintains the full sequence length, while the channel dimension corresponding to all tokens is pruned. The experimental results on LongBench are shown in Table 6. Directly applying channel pruning on the LLaMA-3-8B-Instruct model still preserves excellent performance, even when 70% of the channels are pruned. More importantly, our method demonstrates significant advantages compared to the Think method. This indicates that our method is applicable not only to models with compressed KV cache but also to performing channel pruning directly on the original model.

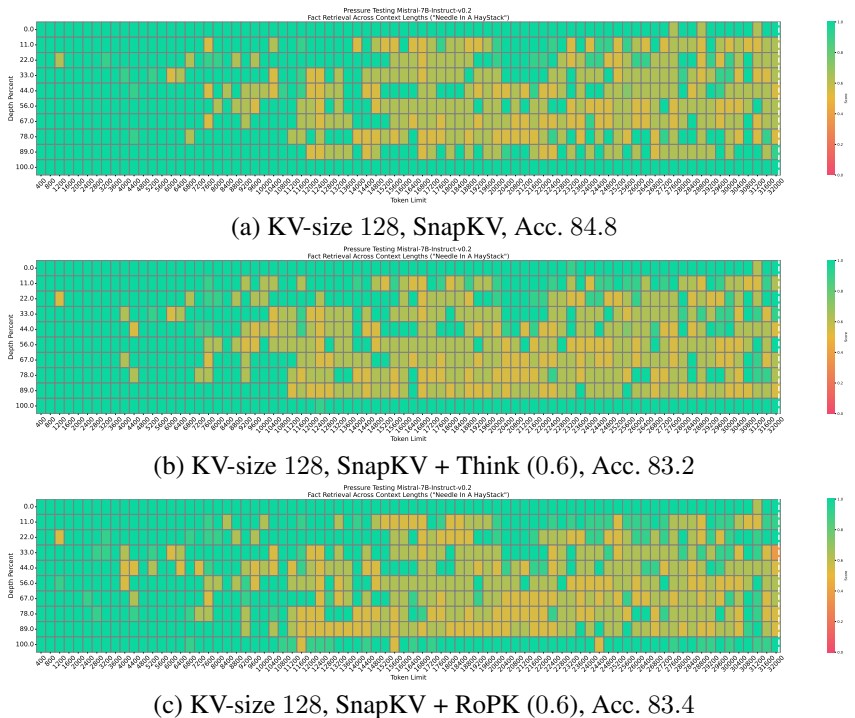

(a) KV-size 128, SnapKV, Acc. 84.8

(b) KV-size 128, SnapKV + Think (0.6), Acc. 83.2

(c) KV-size 128, SnapKV + RoPK (0.6), Acc. 83.4

Figure 4: Experiment results of Mistral-7B-Instruct-v0.2 on Needle-in-a-Haystack benchmark with KV size is set to 128. Think($p$) and RoPK($p$) denote adopting a pruning rate of $p$ for the key cache channels.

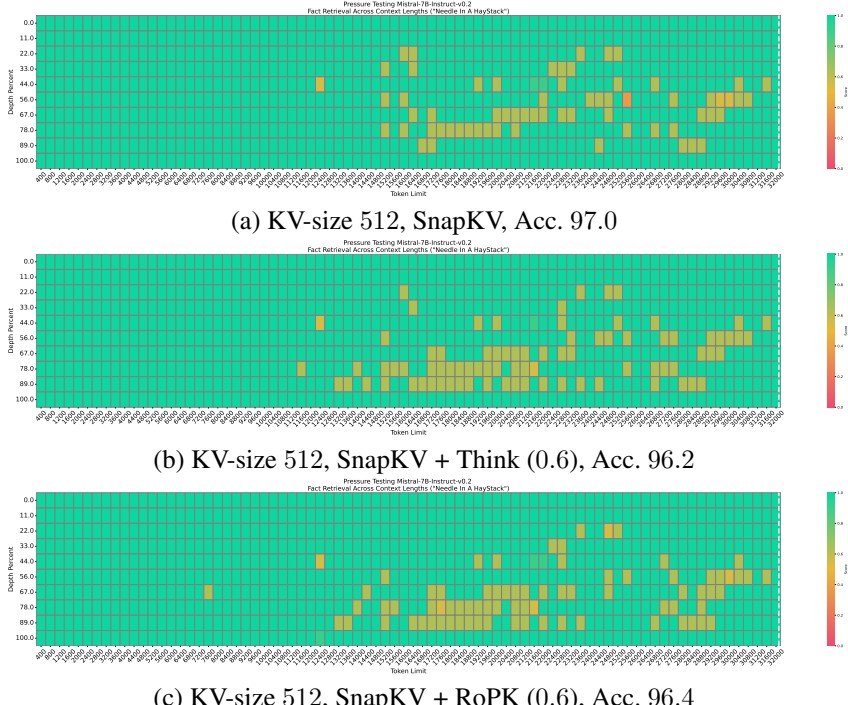

(a) KV-size 512, SnapKV, Acc. 97.0

(b) KV-size 512, SnapKV + Think (0.6), Acc. 96.2

(c) KV-size 512, SnapKV + RoPK (0.6), Acc. 96.4

Figure 5: Experiment results of Mistral-7B-Instruct-v0.2 on Needle-in-a-Haystack benchmark with KV size is set to 512. Think($p$) and RoPK($p$) denote adopting a pruning rate of $p$ for the key cache channels.

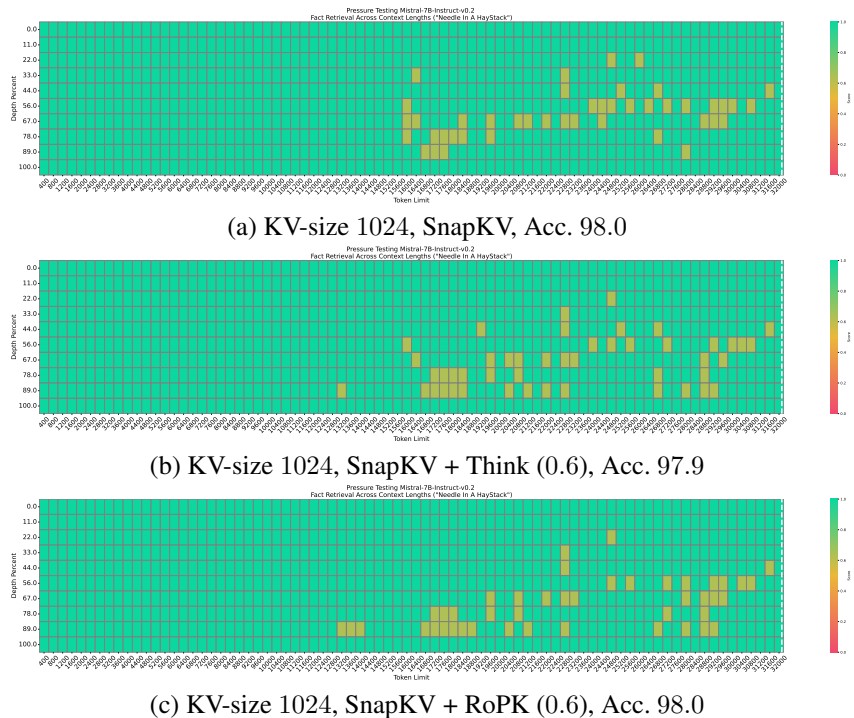

(a) KV-size 1024, SnapKV, Acc. 98.0

(b) KV-size 1024, SnapKV + Think (0.6), Acc. 97.9

(c) KV-size 1024, SnapKV + RoPK (0.6), Acc. 98.0

Figure 6: Experiment results of Mistral-7B-Instruct-v0.2 on Needle-in-a-Haystack benchmark with KV size is set to 1024. Think($p$) and RoPK($p$) denote adopting a pruning rate of $p$ for the key cache channels.

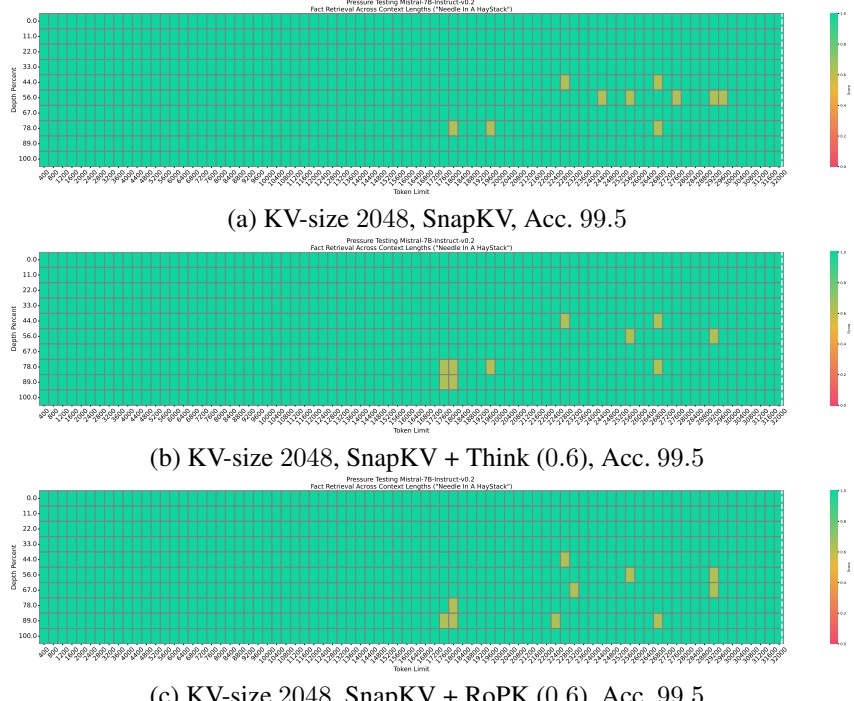

(a) KV-size 2048, SnapKV, Acc. 99.5

(b) KV-size 2048, SnapKV + Think (0.6), Acc. 99.5

(c) KV-size 2048, SnapKV + RoPK (0.6), Acc. 99.5

Figure 7: Experiment results of Mistral-7B-Instruct-v0.2 on Needle-in-a-Haystack benchmark with KV size is set to 2048. Think($p$) and RoPK($p$) denote adopting a pruning rate of $p$ for the key cache channels.

Table 6: LongBench results of directly performing key cache channel pruning on LLaMA-3-8B-Instruct. Think($p$) and RoPK($p$) represent the pruning rate is $p$ for the key cache channels.

| Method | Single-Document QA | | | Multi-Document QA | | | Summarization | | | Few-shot Learning | | | Synthetic | | Code | | Avg. |
|---|---|---|---|---|---|---|---|---|---|---|---|---|---|---|---|---|---|
| | NrtvQA | Qasper | MF-en | HotpotQA | 2WikiMQA | Musique | GovReport | QMSum | MultiNews | TREC | TriviaQA | SAMSum | PCount | PRe | Lcc | RB-P | |
| | | | | | | LLaMA-3-8B-Instruct, KV-size Full | | | | | | | | | | | |
| Vanilla | 25.56 | 32.27 | 39.71 | 43.56 | 35.09 | 21.18 | 28.71 | 23.26 | 26.64 | 73.50 | 90.48 | 42.33 | 4.80 | 69.25 | 59.29 | 54.05 | 41.86 |
| +Think(0.7) | 25.22 | 17.68 | 38.12 | 37.49 | 28.77 | 18.87 | 20.03 | 22.06 | 19.87 | 53.00 | 88.31 | 34.79 | 5.61 | 69.12 | 53.52 | 55.06 | 36.72 |
| **+RoPK(0.7)** | 24.71 | 27.41 | 40.81 | 39.17 | 30.83 | 18.88 | 20.97 | 23.13 | 23.19 | 70.50 | 90.43 | 35.79 | 6.14 | 69.50 | 53.32 | 55.42 | **39.39** |

