# OpenReview forum: "RoPK: A Head-Level Key Cache Channel Pruning Method for Efficient Long-Context LLM Inference"
_ICLR.cc/2026/Conference — Submitted to ICLR 2026_

### Official Review · Reviewer_Ltgs · 2025-10-20

**Soundness:** 3
**Presentation:** 3
**Contribution:** 2
**Rating:** 6
**Confidence:** 3

**Summary:**

This paper proposes RoPK, a head-level key cache channel pruning method for efficient long-context large language model (LLM) inference.
The key idea is to exploit the frequency properties of the Rotary Position Embedding (RoPE) to estimate each attention head’s importance, and allocate different pruning ratios across heads accordingly.
Specifically, the authors define a head importance score based on the Spearman correlation between low-frequency contribution and token distance, and perform frequency-aware structured pruning on the key cache channels of each head.
Experiments on LongBench, RULER, and Needle-in-a-Haystack show that RoPK can prune up to 60–70% of key cache channels with almost no accuracy degradation.

**Strengths:**

1.Presents a novel frequency-aware and head-adaptive channel pruning strategy for long-context LLMs.

2. Provides intuitive and partially theoretical reasoning based on RoPE frequency decomposition.

3. Demonstrates strong empirical performance—up to 60–70% KV compression with minimal accuracy drop.

4. The approach is orthogonal to existing token-level pruning (e.g., SnapKV, H2O) and can be combined with them.

**Weaknesses:**

1. The mapping from head importance to pruning ratio is heuristic; lacks derivation from an explicit optimization objective.

2. No runtime or memory benchmarks—efficiency is shown only in accuracy terms, not in actual throughput or GPU memory savings.

3. Limited compatibility with mainstream inference frameworks (FlashAttention, vLLM, SGLang) that require uniform per-head dimensions.

4. Does not analyze potential stability or generalization issues under different architectures or datasets.

**Questions:**

Is this method compatible with mainstream inference frameworks (FlashAttention, vLLM, SGLang)?

---

### Official Review · Reviewer_GDJe · 2025-10-25

**Soundness:** 2
**Presentation:** 2
**Contribution:** 1
**Rating:** 2
**Confidence:** 5

**Summary:**

This paper addresses the excessive memory overhead of the KV cache in LLMs, which limits inference efficiency and context length. To tackle this, it proposes RoPK, a head-level key cache channel pruning method that allocates non-uniform channel budgets to attention heads based on a new head importance metric derived from RoPE. The authors observe that different RoPE frequencies specialize in modeling different dependency ranges—low frequencies for long-range and high frequencies for short-range and use the correlation between frequency contribution and relative distance to quantify each head’s importance. Important heads are allocated more channels, while less important ones are pruned more aggressively. Experiments on LLaMA-3 and Mistral models show that RoPK achieves up to 60–70% pruning with almost no performance loss, outperforming the Think baseline by around 2% on LongBench.

**Strengths:**

1. The paper targets a genuine bottleneck, KV Cache memory overhead in long-context LLM inference, a problem of high current interest.
2. Experiments span multiple models (LLaMA-3, Mistral) and benchmarks (LongBench, RULER, Needle-in-a-Haystack), demonstrating consistent gains over baselines.

**Weaknesses:**

1. The theoretical formulation of the RoPE frequency importance connection remains heuristic and lacks a rigorous justification of why the correlation metric consistently aligns with head contribution to performance.
2. The estimation procedure for head importance introduces extra computation and potential instability, which could hinder real-world deployment efficiency.
3. KV cache pruning has been extensively explored in many prior works, and compared to these existing studies, this paper’s contribution appears rather incremental.

**Questions:**

Comparisons are primarily limited to Think and token-eviction baselines; further experiments with quantization or low-rank compression methods would strengthen claims of generality and orthogonality.

---

### Official Review · Reviewer_naei · 2025-10-31

**Soundness:** 3
**Presentation:** 2
**Contribution:** 2
**Rating:** 4
**Confidence:** 3

**Summary:**

This paper proposes RoPK, a novel head-level channel pruning method for the LLM kv cache. The core contribution is a new head importance metric derived from rope. This metric quantifies how well an attention head specializes its frequency usage based on relative distance, positing that important heads use low frequencies for long-range and high frequencies for short-range dependencies. Based on this score, RoPK non-uniformly allocates channel budgets to different heads and prunes low-contribution channels within each head. Experiments on LLaMA and Mistral models demonstrate that RoPK significantly compresses the key cache, outperforming prior channel pruning methods on standard long-context benchmarks.

**Strengths:**

- The paper's primary strength is its well-motivated head importance metric. Grounding the pruning strategy in the theoretical properties of RoPE—specifically, frequency specialization for short-versus long-range dependencies is aneffective idea.
- This non-uniform allocation strategy demonstrates clear and significant performance gains over the state-of-the-art channel pruning method (Think) at high compression rates.
- The method's orthogonality to token eviction techniques, which is well-demonstrated (w/ snapkv), further enhances its practical value.
- The experiments are comprehensive.

**Weaknesses:**

- The primary weakness is the unaddressed computational overhead of the head importance metric, which requires sampling query-key pairs and computing correlations. The paper analyzes the sample size but not the full pre-computation cost or its sensitivity across domains.
- Additionally, the framing as purely "key cache" pruning is slightly imprecise; the method also applies the pruning mask $M_k$ to the query matrix $Q_k$ during the attention calculation. The value cache $V_k$ remains uncompressed.
- The paper does not have any wall clock numbers and system implementation. For example, some latency/throughputs. Almost everything is about accuracy and does not consider the system side implementation. I recommend the authors to include this part.

**Questions:**

- Please clarify the computational overhead of the head importance metric calculation. Is this a one-time, offline cost per model, or must it be re-calibrated for different tasks?
- The method prunes both $Q_k$ and $K_k$ via the mask $M_k$. Why is the value cache $V_k$ not also pruned along the same channel dimensions? What would be the performance impact of such an extension?
- Could you include the system side experiments?

---

### Official Review · Reviewer_1FuN · 2025-10-31

**Soundness:** 2
**Presentation:** 3
**Contribution:** 2
**Rating:** 4
**Confidence:** 4

**Summary:**

This paper proposes RoPK, a head-level key cache channel pruning method for long-context LLM inference. RoPK introduces a head importance metric based on RoPE frequency, allocating more budget to important heads and pruning less important channels within each head. Evaluation on several benchmarks shows that RoPK can achieve minimal performance loss.

**Strengths:**

- The paper is well-motivated, exploring the underexplored channel pruning for LLM KV cache acceleration. And the intuition of developing a channel allocation method based on RoPE makes sense, with detailed analysis in section 4.1, 4.2.

- RoPK is rigorously evaluated on LongBench, RULER, and Needle-in-a-Haystack, demonstrating robustness across popular long-context benchmarks.

**Weaknesses:**

- Evaluation: Why does the evaluation integrate RoPK with SnapKV? Why not directly evaluate its performance? And efficiency evaluation is missing (e.g. tps, latency). This is important for a KV cache pruning paper, since it is designed to accelerate the inference speed; evaluation only on performance is not enough.

- Limited Model : Experiments are conducted only on LLaMA-3-8B-Instruct and Mistral-7B-Instruct-v0.2 (Why not  LLaMA-3.1-8B-Instruct/ Llama-3-8B-1024K?). These models are outdated with a small context window and poor long-context ability compared to SOTA models. Testing on more architectures and newer models would strengthen generalizability. (e.g. Phi-3-Mini-128K, Qwen2.5/3, etc).

- Missing Baselines: Some more recent baselines are missing, such as ShadowKV, Duoattention.

**Questions:**

Why only use a 4K context length setting in RULER? Performance on much longer contexts (e.g., 128K on RULER) remains unverified, which is key for long-context models.

---

### Meta-Review · Area_Chair_TYx7 · 2026-01-04

**Summary:**

The review identifies critical flaws in this KV cache pruning paper: it lacks standalone RoPK performance tests and key efficiency metrics (TPS, latency) while overprioritizing accuracy despite its inference acceleration goal. Experiments rely on outdated small-context models, excluding newer/SOTA variants and weakening generalizability. Recent baselines are omitted; the "key-only pruning" claim is imprecise, and head importance calculation overhead is unaddressed. Its heuristic theoretical foundation yields incremental contributions, with no system benchmarks, poor framework compatibility, and unanalyzed stability/generalization.

The authors did not provide a rebuttal to address any concerns. Therefore, the suggested decision is reject.

**Reviewer Concerns:**

The authors did not provide a rebuttal. Therefore, all concerns are not addressed.

**Reviewer Scores:**

The reviewers would maintain or lower their scores as the lack of rebuttal.

---

### Decision · Program_Chairs · 2026-01-26

Reject